# Incomplete Kawasaki Disease with Peripheral Facial Nerve Palsy and Lung Nodules: A Case Report and Literature Review

**DOI:** 10.3390/children10040679

**Published:** 2023-04-03

**Authors:** Marco Maglione, Annalisa Barlabà, Michela Grieco, Rosaria Cosimi, Giangiacomo Di Nardo, Giovanni Maria Di Marco, Monica Gelzo, Giuseppe Castaldo, Celeste Tucci, Raffaella Margherita Iodice, Maria Concetta Lonardo, Vincenzo Tipo, Antonietta Giannattasio

**Affiliations:** 1Pediatric Emergency Unit, Santobono-Pausilipon Children’s Hospital, 80129 Naples, Italy; 2Department of Woman, Child and of General and Specialized Surgery, Università degli Studi della Campania L. Vanvitelli, 81100 Naples, Italy; 3Department of Pediatric Cardiology, Santobono-Pausilipon Children’s Hospital, 80129 Naples, Italy; 4CEINGE-Biotecnologie Avanzate, Scarl, 80131 Naples, Italy; 5Dipartimento di Medicina Molecolare e Biotecnologie Mediche, Università di Napoli Federico II, 80138 Naples, Italy; 6Department of Neuroscience, Pediatric Neurology, Santobono-Pausilipon Children’s Hospital, 80129 Naples, Italy

**Keywords:** infants, coronary artery aneurysm, facial nerve palsy, Kawasaki disease

## Abstract

The diagnosis of Kawasaki disease (KD) is challenging and often delayed mainly in case of young infants and in presence of an incomplete disease and atypical features. Facial nerve palsy is one of the rare neurologic symptoms of KD, associated with a higher incidence of coronary arteries lesions and may be an indicator of a more severe disease. Here, we describe a case of lower motor neuron facial nerve palsy complicating KD and perform an extensive literature review to better characterize clinical features and treatment of patients with KD-associated facial nerve palsy. The patient was diagnosed at the sixth day of disease and presented extensive coronary artery lesions. A prompt treatment with intravenous immunoglobulins, aspirin and steroids obtained a good clinical and laboratory response, with resolution of facial nerve palsy and improvement of coronary lesions. The incidence of facial nerve palsy is 0.9–1.3%; it is often unilateral, transient, more frequent on the left and seemingly associated with coronary impairment. Our literature review showed coronary artery involvement in the majority of reported cases (27/35, 77%) of KD with facial nerve palsy. Unexplained facial nerve palsy in young children with a prolonged febrile illness should prompt consideration of echocardiography to exclude KD and start the appropriate treatment.

## 1. Introduction

Kawasaki disease (KD) is an acute, self-limited heterogeneous disease of unknown etiology that predominantly affects infants and children less than 5 years of age [1]. No diagnostic test is currently available for this condition, and typical or classic KD is diagnosed on a clinical basis in presence of specific criteria and after other similar clinical entities have been excluded. Classic KD is characterized by the presence of ≥5 days of fever and ≥4 of the following main clinical features: bilateral non-exudative conjunctivitis, erythema of lips and oral mucosa, changes in the extremities, skin rash, and cervical lymphadenopathy [1]. Patients who do not fulfill complete diagnostic criteria for KD are often referred to as atypical or incomplete KD [1,2].

The most feared complication of KD is the development of coronary artery lesions (CALs) that make KD the most common cause of acquired cardiac disorder in children [3]. In other cases, patients present clinical manifestations not included in the diagnostic criteria such as myocarditis, pericarditis, peri-bronchial and interstitial infiltrates on chest radiography, abdominal pain, behavioral changes and irritability, aseptic meningitis, and peripheral facial nerve palsy (FNP) [2]. This last manifestation is rare and nearly all available literature is limited to isolated case reports. We describe a young infant with KD presenting with FNP, lung nodules and severe CALs. We also performed an extensive literature review to better characterize clinical features and treatment of patients with KD-associated FNP.

## 2. Case Presentation

A 4-month-old boy presented with a 6-day history of fever treated with antibiotics. On day 4 of fever, bilateral conjunctival injection and a confluent, erythematous and papular rash of distal extremities appeared. As fever persisted, and given the abnormal facial movements noted by the parents, he presented to the Pediatric Emergency Department. At hospital admission, physical examination revealed impaired facial expression when he cried and food refusal. Neurologic assessment confirmed a lower motor-neuron-type palsy of the right facial nerve with no other neurological deficit (Figure 1).

A head computed tomography (CT) showed no focal abnormalities. Brain magnetic resonance imaging and lumbar puncture were not performed. Blood tests at admission (Table 1 and Figure 2) showed leukocytosis, normocytic normochromic anemia, increased platelets count and C-reactive protein (CRP), and a slightly increased procalcitonin (PCT, 0.82 ng/mL, reference value < 0.5). He also presented elevated serum ferritin and reduced serum albumin levels (21 g/L). The most important markers of cardiac injury (cardiac myoglobin, creatine kinase-MB and brain natriuretic peptide, BNP) were within the normal range.

Serum cytokines analysis showed an increased tumor necrosis factor alpha (TNF-α), and a slight increase in interleukin (IL)-10, IL-1-β and IL-17A (Table 2).

Blood, throat, stool, and urine cultures were negative. Polymerase chain reaction for bacterial and viral pathogens on throat swab showed positivity for human rhinovirus/enterovirus and Parainfluenza virus 1. Chest X-ray was unremarkable. Serologic tests were negative for severe acute respiratory syndrome coronavirus 2 (SARS-CoV-2), herpes simplex virus (HSV) types 1 and 2, enterovirus, adenovirus, Mycoplasma pneumoniae, Epstein–Barr virus and cytomegalovirus. An empiric antibiotic therapy with ceftazidime was started, while the laboratory results were pending.

Based on patient’s medical history and clinical and laboratory findings, an incomplete KD was suspected. A cardiac assessment with electrocardiogram (ECG) and echocardiography was performed. At admission, echocardiography showed medium-size fusiform aneurysms of the right CA (4 mm, z-score 9), medium-size dilatation of the proximal right CA (3.3 mm, z-score 6) and small-size dilatation of the left CA (2.8 mm, z-score 3.9) (Figure 3).

The diagnosis of KD was then confirmed. Therapy with intravenous immunoglobulin (IVIG 2 g/kg) associated to corticosteroids (intravenous methylprednisolone 30 mg/kg/day for three days) and oral acetylsalicylic acid (ASA, 50 mg/kg/day) was started. Resolution of fever and irritability was observed within 24 h. Laboratory findings showed a progressive decrease of CRP, ferritin, leukocytes, and platelets (Table 1 and Figure 2). On day 4 of hospitalization, he underwent a cardiac CT that showed a typical “beaded appearance” of right (dominant) CA due to aneurysmatic dilatations in the absence of thrombi. The proximal tract of right CA showed a 7 mm-long aneurysmatic dilatation with maximum diameters of 4 × 3 mm, while the medium tract showed a 7 mm-long aneurysmatic dilatation with maximum diameters of 5 × 4 mm. The proximal tract of the conal branch also showed a focal saccular dilatation (diameters of 3 × 2 mm). At its origin, the posterolateral branch showed a focal aneurysmatic dilatation (maximum diameter, 3 mm); the left anterior descending artery branch showed a focal aneurysmatic dilatation (diameters 3 × 3.5 mm). Two lung nodules (10 × 9 mm and 4 mm in diameter, respectively) located in the left upper lobe were also detected (Figure 4).

On the basis of the severity of the coronary lesions and the of the z-score value, 4 days after admission, clopidogrel (1.3 mg/day) was started and ASA was stopped. Steroids were tapered and then switched to oral route.

The patient was discharged after 23 days with a strict cardiological follow-up. His left facial weakness was unappreciable at discharge. Three months after discharge, echocardiographic evaluation showed an improvement of CA lesions with a mild dilatation (2.8 mm, z-score 5) of the right CA in its median tract, with complete resolution of the lesions of the proximal tract of right CA and of left CA.

## 3. Discussion

Clinical diagnosis of KD in infants below 1 year of age, and even more below 6 months, can be very challenging since patients often do not present the typical signs and symptoms, and the risk of diagnostic delay is therefore greater. Mastrangelo and coworkers described clinical and laboratory findings in infants below 12 months of age with KD and reported a common presentation with incomplete disease [4]. In addition, KD in early infancy is associated with an increased risk of cardiac involvement that is detected in about 60% of patients in both complete and incomplete KD [4]. Coronary artery aneurysms (CAA) are detected in 50% of patients, most of whom have an incomplete form of KD [4]. Similarly, Salgado reported a higher incidence of dilated or aneurysmal CA in infants <6 months of age compared with those older than 6 months (43.4% vs. 19.5%), even when treatment was started within the first 10 days after fever onset [5].

Our patient was only 4 months old, and we made a diagnosis of incomplete KD because of fever persisting for more than 5 days associated with skin polymorphous rash, conjunctivitis, and changes in the extremities. The diagnosis was strengthened by several supporting laboratory findings, namely thrombocytosis, anemia, hypoalbuminemia and raised inflammatory markers. In addition, the child developed a left FNP and cardiac damage. FNP is a rare neurological manifestation of KD [6]. The first case of FNP in KD was reported in 1974 by Murayama in a 4-month-old male [7]. To date, about 50 cases of infants with KD-associated FNP have been described. Table 3 summarizes the clinical features, treatment, and prognosis of 35 patients with KD and FNP [8,9,10,11,12,13,14,15,16,17,18,19,20,21,22,23,24,25,26,27,28,29].

The incidence of FNP in KD is 0.9–1.3% [29], and it is often unilateral, transient with a spontaneous resolution, more frequent on the left side and seemingly associated with coronary impairment [20]. Although FNP may develop at any time of KD, its onset has been estimated to occur, on average, at the 16th day of illness [22]. In our patient, FNP appeared earlier in comparison with most literature reports (Table 3). This early onset may be due to the severe inflammation observed in our child, as demonstrated by a significant increase in inflammatory markers at admission (CRP, ferritin, platelet, and cytokine levels).

A study conducted by Poon et al. reported 28 patients with FNP as a complication of KD in patients with 3 to 25 months of age, mainly occurring in females, and lasting from 2 days to 3 months, more frequent on the left side and with associated CAA at least in half of the patients [13]. A prospective study by Alves and coworkers summed KD complications and its rare clinical features [20]. He reported an incidence of central nervous system (CNS) involvement in KD ranging from 1.1% to 3.7%: reported manifestations included ataxia, FNP and sensorineural auditory loss [20]. In this review, the only patient showing FNP experienced it during the subacute phase and presented concomitant ataxia, hearing loss and a small left CAA [20]. In a recently published observational study including nine KD patients presenting FNP, its duration ranged from 10 to 130 days. Only one patient showed bilateral FNP [29].

A strict association between presentation with FNP and CAA has been reported, as coronary involvement is present in about two thirds of cases [22]. In the current analysis, CA involvement was described in most cases (27/35, 77%) of KD with FNP. A possible explanation of this strict association may be that the presence of incomplete KD and FNP lead to a delayed diagnosis and thus to a delayed treatment of the condition. Wright et al. described a child with a missed diagnosis of KD during the acute phase because of attribution of a neurologic sign to a possible concurrent otitis media [16]. Nevertheless, in another study comparing patients with FNP to matched KD patients without FNP, the incidence of CALs in KD patients with FNP was much higher than that of the matched KD patients without FNP [29]. These data indicated that KD patients with FNP appeared to have a real higher incidence of CALs.

In our patient, cardiac assessment was based on basal and serial echocardiography and cardiac CT performed on the 4th day after hospitalization. Cardiac CT (CCT) is superior to echocardiography in CA visualization and characterization, especially for distal CA segments [30]. In our case, CT scans allowed to exclude the presence of CA thrombosis or stenosis and to better characterize CA involvement. Furthermore, it identified two lung nodules that were not revealed by chest X-ray. It is important to note that our patient had no associated respiratory symptoms. Lung lesions in KD remain a diagnostic challenge. Pulmonary nodules have an inflammatory nature and rapidly regressed with the standard KD treatment [31,32]. Given the absence of respiratory manifestations and the reported spontaneous resolution of nodules in the literature, we did not perform a follow-up chest CT scan. Indeed, clinical improvement with proper treatment of KD allows to avoid serial CT scans in order to reduce both radiation exposure and frequency of general anesthesia, which is needed for image acquisition in young children.

As for the pathogenesis of FNP in KD, several etiologies have been hypothesized. It is likely that ischemic vasculitis of the arteries supplying the facial nerve contributes to FNP [11] together with an inflammatory process of the facial nerve itself [22,26]. This hypothesis may explain the positive effect of IVIG on FNP, likely due to a modulating effect on the synthesis and release of proinflammatory cytokines [33]. In a recent study, FNP occurred in 44.4% of patients despite the use of IVIG [29]. It seems that a prompt therapy with IVIG may resolve FNP more quickly, but it is not able to decrease the excessive inflammatory activity in these patients [22].

The current treatment for KD is a high dose (2 g/kg) of IVIG, administered within the first 10 days after disease onset [34,35]. Additionally, high-dose aspirin is advised by the American Heart Association. Most patients show a quick clinical response after IVIG therapy, yet approximately 10–20% of all patients do not respond well or have recurrent fever within 36–48 h after treatment [36]. Age of <6 months has been reported to be an important variable in predicting IVIG resistance [37,38]. In presence of high-risk patients with KD (children less than 12 months or children having CRP higher than 200 mg/l, severe anemia at disease onset, albumin level below 2.5 g/dl, liver disease, overt coronary artery aneurysms, macrophage activation syndrome or septic shock), Italian guidelines recommended high dose of intravenous methylprednisolone and ASA at anti-platelet dose [34]. Treatment of KD with FNP is less standardized. In a recent review including nine patients, all children received IVIG (2 g/kg), ASA (30–50 mg/kg/day) and short-term dexamethasone [29]. Our patient was considered a high-risk infant and was treated with IVIG, high dose of steroids and ASA. Treatment obtained a prompt clinical and laboratory response, while CA damage improved at a medium-term follow up. As expected, CAAs require more time to recover than CA dilation. In presence of CAA, a frequent and long-term follow-up echocardiography is mandatory.

## 4. Conclusions

This case highlights the importance to bear in mind KD diagnosis in any child with prolonged unexplained high fever and FNP. FNP is an unusual manifestation of KD but its presence may be associated to a high risk of severe CAA. For this reason, it is important to not delay the diagnosis of KD in order to start a prompt treatment. Despite its incomplete presentation, the diagnosis of KD was not delayed in our child, and treatment was started a few hours after hospital admission. Unlike CAA, the paralysis normally evolves towards a complete resolution without sequelae, as in the described child. A long-term echocardiography follow-up is required in patients with KD and CAAs.

## Figures and Tables

**Figure 1 children-10-00679-f001:**
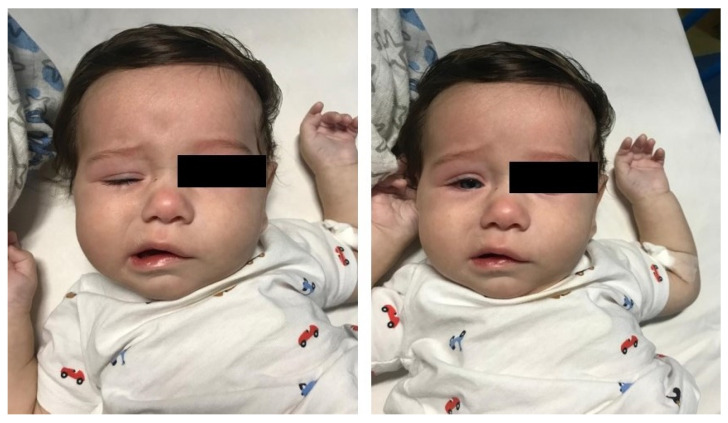
Right facial nerve palsy in the 4-month-old infant. These clinical features were observed at hospital admission (closed eyes on the left; open eyes on the right).

**Figure 2 children-10-00679-f002:**
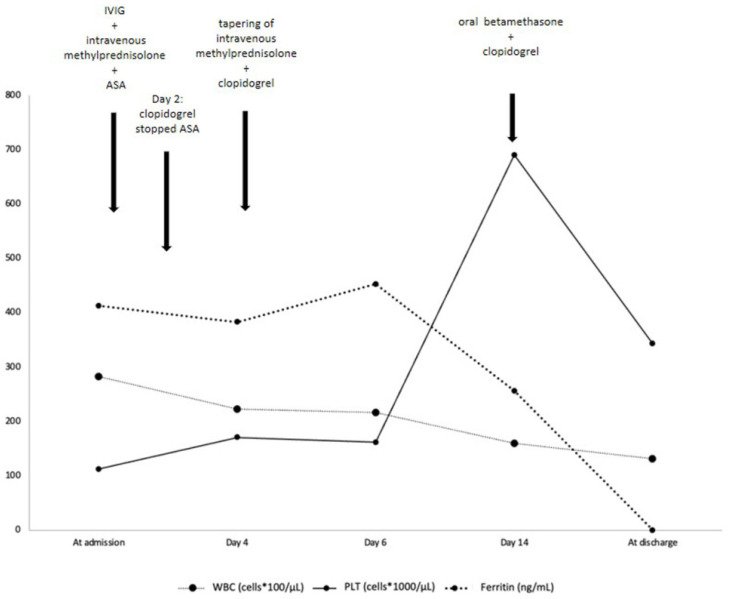
Timeline evolution of most relevant laboratory findings and treatment. IVIG = intravenous immunoglobulin; ASA = acetylsalicylic acid; WBC= white blood cells; PLT = platelets. Ferritin (ng/mL) reference value < 320.

**Figure 3 children-10-00679-f003:**
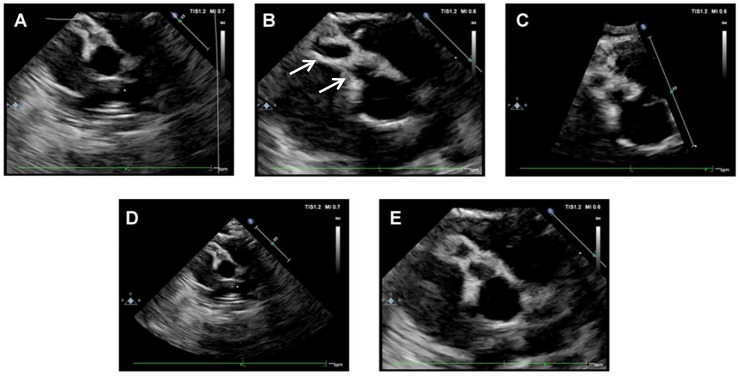
Echocardiographic scans at hospital admission. (**A**–**D**): medium-size fusiform aneurysms of right CA (beaded appearance). In Figure (**B**), arrows indicated aneurysms of right CA. (**E**): medium-size dilatation of proximal right CA.

**Figure 4 children-10-00679-f004:**
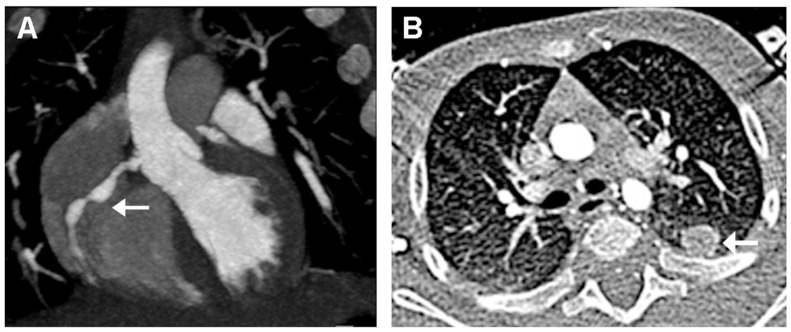
Cardiac CT performed 4 days after hospital admission. (**A**): “Beaded appearance” (indivated by arrow) of the aneurysmatic dilatations of the right CA. (**B**): Lung nodule (indicated by arrow) in the left upper lobe.

**Table 1 children-10-00679-t001:** Laboratory findings during hospital stay.

	At Admission	Day 2	Day 4	Day 6	Day 14	At Discharge
Hb (g/dl)	8.6		9.1	10.9	11.4	11.8
CRP (mg/L) (reference value < 5)	111.31		8.48	1.77	0.30	0.44
ESR (mm/h)	54				23	
AST (U/L) (reference value < 48)	20	18	27	42	30	36
ALT (U/L) (reference value < 50)	14	14	22	44	27	17
Ferritin (ng/mL) (reference value < 320)	412.9		383.2	453.30	256.8	
Troponin (ng/L) (reference value < 14)	9	6	6	9	7	
BNP (pg/mL) (reference value < 100)	48.2		28.4	8.3	<5	

Hb = hemoglobin; CRP = C-reactive protein; ESR = erythrocyte sedimentation rate; AST = aspartate aminotransferase; ALT = alanine aminotransferase; BNP = b-type natriuretic peptide.

**Table 2 children-10-00679-t002:** Serum cytokine analysis.

Cytokine	Patient’s Value	Reference Value
IFN-γ (pg/mL)	0.73	0.54–2.72
IL-1-β (pg/mL)	0.58	<0.16
IL-4 (pg/mL)	0.35	<0.5
IL-6 (pg/mL)	3.54	0.76–6.38
IL-10 (pg/mL)	4.64	1.77–3.76
IL-12p70 (pg/mL)	0.78	0.60–7.96
IL-17A (pg/mL)	1.54	<1.05
TNF-α (pg/mL)	16.9	7.78–12.2

IFN = interferon; IL = interleukin; TNF = tumor necrosis factor.

**Table 3 children-10-00679-t003:** Published articles including children with KD and concomitant facial nerve palsy.

Author	Article Type	Patient’s Age (Months)	No. of Included Patients	FNP Onset (Day of Illness)	Coronary Involvement	Therapy	Outcome
Terasawa, 1983 [8]	Case report	12	3	22	No	ASA (30 mg/kg)	Full recovery of FNP within 2 weeks
Hattori, 1987 [9]	Case report	6; 9	2	16; 19	No	ASA (50 and 80 mg/kg/d, respectively)	Full recovery of FNP within 1 and 2 months, respectively
Park, 1991 [10]	Case report	4	1	17	Not reported	IVIG (400 mg/kg/d for 5 days), ASA (100 mg/kg/d)	Full recovery of FNP within 40 days
Bushara, 1997 [11]	Case report	3	1	22	Aneurysmal dilatation of both right and left CAs	IVIG (2 g/kg), ASA (100 mg/kg/d), heparin	Improvement of CAA
McDonald, 1998 [12]	Case report	3 and 2 weeks	1	11	Aneurysms of the left anterior descending and circumflex CAs	Not reported	Not reported
Poon, 2000 [13]	Case report	24	1	>15 days	No	IVIG (2 g/kg) and ASA (30 mg/kg/d)	Full recovery of FNP within 1 month
Biezeveld, 2002 [14]	Case report	156	1	12	Right CAA	IVIG (2 g/kg), ASA (80 mg/kg/d)	Resolution of CAA within 1 month
Larralde, 2003 [15]	Case report	5	1	>11	Right CAA and fusiform enlargement of the left CA	IVIG (2 g/kg), ASA (100 mg/kg/d)	Complete recovery
Wright, 2008 [16]	Case report	3; 7	2	6; not reported	“Beaded” appearance of the right CA, ectasia of proximal right CA and left anterior descending CA, severe bilateral axillary artery involvement	IVIG (2 g/kg), ASA (75 mg/kg/d), warfarin, ASA (started after the acute phase of KD)	Minimal residual dilatation of the right CA and slight ectasia of the other CAs at the 6th month post discharge; stable at 9-year follow-up
Li, 2008 [17]	Case report	-	2	-	CAAs	IVIG	Complete resolution of CAA in one case; persistence of CAA in the other case
Lim, 2009 [18]	Case report	72	1	10	Aneurysm of the left and right CAs, left anterior descending CA, left circumflex CA; giant left anterior descending CA aneurysm on day 13 of illness	IVIG (2 g/kg, 2 doses), ASA (100 mg/kg/d), warfarin and prednisolone (1 mg/kg/d)	No symptoms of myocardial ischemia on exertion
Kaur, 2010 [19]	Case report	2	1	12	Dilatation of right and left CAs and left anterior descending CA	IVIG (2 g/kg), ASA (75 mg/kg/d)	Resolution of FNP within 2 days
Alves, 2011 [20]	Prospective study	38 (mean age)	115, FNP in one case	26	Small left CAA	IVIG (2 g/kg)	FNP improvement within 30 days
Khubchandani, 2014 [21]	Case report	36	1	27	Diffuse dilatation of all CAs and aneurysm of left anterior descending and proximal right CA	IVIG (2 g/kg, 2 doses)	Full recovery of FNP within 3 weeks
Stowe, 2015 [22]	Case report	15	1	6	Right CA dilatation	IVIG (2 g/kg), high-dose ASA, i.v. methylprednisolone (30 mg/kg/d)	Full recovery
Rodriguez-Gonzalez, 2018 [23]	Case report	5	1	12	Aneurysms of the left and right main CAs, left anterior descending CA, and left circumflex CA	IVIG (2 g/kg), ASA (100 mg/kg/d), steroids	Full recovery of FNP and improvement of CAA
Orgun, 2018 [24]	Case report	4	1	7	Left CA, proximal right CA and CA ectasia at all segments, saccular aneurysm in proximal right CA	IVIG (2 g/kg/12 h), ASA (80 mg/kg/d), enalapril (0.1 mg/kg/d), subcutaneous enoxaparin	FNP recovered 7 days after IVIG treatment, resolution of CAAs, minimal improvement of CA ectasia
Yuan, 2019 [25]	Case report	3	1	8	Left CA dilatation	IVIG (2 g/kg/16 h), ASA (50 mg/kg/d)	Full recovery
Zhang, 2019 [26]	Case report	6	1	6	Bilateral dilatation of CAs and CAA	IVIG (2 g/kg), ASA (four doses of 30–50 mg/kg/d)	Persistence of FNP during the 18-month follow-up
Yu, 2019 [27]	Case report	7	1	14	Dilatation of all CAs, in addition to aneurysms of the middle of the right and left CAs	IVIG (2 g/kg), high-dose ASA	Partial FNP improvement during the follow-up period
Peña-Juárez, 2021 [28]	Case report	9	1	29	Bilateral giant CAAs	IVIG (2 g/kg), high-dose ASA, methylprednisolone (30 mg/kg)	-
Chen, 2021 [29]	Retrospective observational study	88.9% of patients <24 and 55.6% <12	9	10 (median time)	In 8 out of 9 patients: CAA in 4 cases and CA dilatation in 4 cases	IVIG (2 g/kg), ASA (30–50 mg/kg/d), short-term dexamethasone	Full remission of CALs in an average time of 66 days

FNP = facial nerve palsy; CAs = coronary arteries; IVIG = intravenous immunoglobulin; ASA = acetylsalicylic acid; CAA = coronary artery aneurysm; KD = Kawasaki disease.

## Data Availability

Not applicable.

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
