# Peer review of "Incomplete Kawasaki Disease with Peripheral Facial Nerve Palsy and Lung Nodules: A Case Report and Literature Review"

_children, 2023, doi:10.3390/children10040679_

Round 1
Reviewer 1 Report
It is a good manuscript, I suggest small changes.
Incomplete Kawasaki disease with peripheral facial nerve palsy and lung nodules: a case report and literature review
I have reviewed the case, it is well-written and comprehensive.
In the abstract, you must add the purpose of the study.
Table 1 you can adjust values and explain better Positive test (HUS) . I suggest creating a graphic series of laboratory parameters, with values from all the hospitalization periods expressed and you can compare better.
A time-line evolution and treatment is also a good idea, for results.
Fig 2 B, do you have a better one? And please add arrows, for a better understanding.
Add the imagistic in dinamics, please.
I suggested that you point out the particularity of your case in the conclusions and reformulate it. congratulations for the case and the article
Best regards
Author Response
In the abstract, you must add the purpose of the study.
Aim of the study has been added in the Abstract, as suggested (“Here we describe a case of lower motor neuron facial nerve palsy complicating KD and we performed an extensive literature review to better characterize clinical features and treatment of patients with KD-associated facial nerve palsy”).
Table 1 you can adjust values and explain better Positive test (HUS) . I suggest creating a graphic series of laboratory parameters, with values from all the hospitalization periods expressed and you can compare better. A time-line evolution and treatment is also a good idea, for results.
We thanks the reviewer for useful suggestion. The most relevant laboratory results (WBC, plt and ferritin values) have been removed from Table 1 and described in a time-line figure (Figure 2) including treatmentsstarted or stopped.
Fig 2 B, do you have a better one? And please add arrows, for a better understanding.
Arrows have been added, as suggested.
Add the imagistic in dinamics, please.
A picture comparing right FNP features at hospital admission with closed and open eyes has been added (Figure 1).
I suggested that you point out the particularity of your case in the conclusions and reformulate it.
Conclusions have been reformulated, as suggested.
Reviewer 2 Report
The authors reported the case of a 4-month-old boy with Kawasaki disease who concurred with facial nerve palsy. Unfortunately, he was complicated with coronary arterial aneurysms in spite of appropriate diagnosis and treatments. As the authors described, FNP associated with KD indicates the risk of development of CAL. Although the issues about cardiac involvements were well described, but the issues about FNP was unsatisfied. I think that the authors should strengthen the neurological issues.
1. The authors have to describe more clinical courses of FNP in detail. It should be reported whether FNP persisted at discharge or at the last follow-up.
2. Did you perform brain MRI or lumber puncture test?
Author Response
- The authors have to describe more clinical courses of FNP in detail. It should be reported whether FNP persisted at discharge or at the last follow-up.
FNP was completely resolved at discharge.This is reported in the last paragraph of case presentation (page 4:” His left facial weakness was unappreciable at discharge”).
- Did you perform brain MRI or lumber puncture test?
Brain MRI and lumbar puncture were not performed in our case. This data has been added in Case presentation.